# Polyurethanes as New Excipients in Nail Therapeutics

**DOI:** 10.3390/pharmaceutics10040276

**Published:** 2018-12-13

**Authors:** Barbara S. Gregorí Valdes, Ana Paula Serro, Joana Marto, Rui Galhano dos Santos, Elena Cutrín Gómez, Francisco J. Otero-Espinar, João Moura Bordado, Helena Margarida Ribeiro

**Affiliations:** 1Research Institute for Medicine (iMed.ULisboa), Faculty of Pharmacy, Universidade de Lisboa, 1649-003 Lisboa, Portugal; azurina2628@gmail.com (B.S.G.V.); jmmarto@ff.ulisboa.pt (J.M.); 2Centre for Natural Resources and the Environment (Cerena), Instituto Superior Técnico, Universidade de Lisboa, 1049-001Lisboa, Portugal; rui.galhano@ist.utl.pt (R.G.d.S.); jcbordado@ist.utl.pt (J.M.B.); 3Centro de Investigação Interdisciplinar Egas Moniz (CiiEM), Instituto Superior de Ciências da Saúde Egas Moniz, 2829-511 Caparica, Portugal; anapaula.serro@ist.utl.pt; 4Centro de Química Estrutural (CQE), Instituto Superior Técnico, Universidade de Lisboa, 1049-001Lisboa, Portugal; 5Department of Pharmacy and Pharmaceutical Technology Santiago de Compostela, University of Santiago de Compostela, 15782 Santiago de Compostela, Spain; elena.cutrin@usc.es (E.C.G.); francisco.otero@usc.es (F.J.O.-E.)

**Keywords:** nail lacquers, polyurethane, onychomycosis, topical treatment, terbinafine hydrochloride, ciclopirox olamine

## Abstract

Onychomycosis affects about 15% of the population. This disease causes physical and psychosocial discomfort to infected patients. Topical treatment (creams, solutions, gels, colloidal carriers, and nail lacquers) is usually the most commonly required due to the high toxicity of oral drugs. Currently, the most common topical formulations (creams and lotions) present a low drug delivery to the nail infection. Nail lacquers appear to increase drug delivery and simultaneously improve the effectiveness of treatment with increased patient compliance. These formulations leave a polymer film on the nail plate after solvent evaporation. The duration of the film residence in the nail constitutes an important property of nail lacquer formulation. In this study, a polyurethane polymer was used to delivery antifungals drugs, such as terbinafine hydrochloride (TH) and ciclopirox olamine (CPX) and the influence of its concentration on the properties of nail lacquer formulations was assessed. The nail lacquer containing the lowest polymer concentration (10%) was the most effective regarding the *in vitro* release, permeation, and antifungal activity. It has also been demonstrated that the application of PU-based nail lacquer improves the nail plate, making it smooth and uniform and reduces the porosity contributing to the greater effectiveness of these vehicles. To conclude, the use of polyurethane in nail formulations is promising for nail therapeutics.

## 1. Introduction

Onychomycosis is a nail fungal infection caused by dermatophytes, non-dermatophytes, and yeast species [1]. Candida species have a high incidence in fingernail infection, present in as many as 75% of cases, and are more prevalent than dermatophytes [2]. In contrast, the incidence of yeasts in toenail infections is much lower; approximately 2–10% cases. Infection attributed to non-dermatophytes are estimated to be 2–65% of cases although higher rates, 15% have been reported [3].

The untreated onychomycosis may worsen, spread to other uninfected locations (other nails or to the surrounding skin) or infect other patients [4].

Nail keratin is an impermeable structure, thus restricting drug access to the organisms causing onychomycosis [5]. The development of therapeutic transungual drug delivery is urgent for patients that are affected by such infections.

Topical treatment would be of special interest for immunosuppressed, diabetic, and elderly patients who suffer from chronic pathologies and follow long-term drug therapies and among whom the prevalence of onychomycosis is higher [5,6]. Despite an active interest in the method aimed at improving the efficacy of such formulations, the permeation study is justified by recent research reports [7,8,9,10].

Many antifungal formulations have been developed. Some of them are related to ciclopirox olamine (CPX) and terbinafine hydrochloride (TH).

CPX is used in nail topical formulations, has been marketed worldwide for about 25 years and is available in different pharmaceutical preparations: creams (Selergo^®^, Mycoster^®^, 1%), lotion (Mycoster^®^, 10 mg/L) [11], gel (Loprox^®^, 0.77%) [12], and solutions, such as Penlac^®^, Batrafen^®^, Mycoster^®^, Ciclopoli^®^, Ony-Tec^®^ and RejuveNail^®^ containing 8% of active substance. They present butyl monoester of poly methyl vinyl ether/maleic acid [5] or hydroxy propyl chitosan as film formers [13]. CPX presents a minimum inhibitory concentration value for *Candida albicans* and *Aspergillus species* between 0.13–4 µg/mL [14,15].

Alternatively, TH is used in systemic and topical formulations (Lamisil^®^ cream, 1%, Mycova^®^ nail coat, 10%, TDT 067 liquid spray, 15 mg/mL and P-3058, hydroxypropyl chitosan-based, 5%) [5,16]. These nail formulations use dodecyl 2-(*N*,*N*-dimethylamino)-propionate hydrochloride, soybean phosphatidylcholine and hydroxypropyl chitosan, respectively [5]. TH, an allylamine derivative, represents the most effective antimycotic drug, presenting a minimum inhibitory concentration against dermatophytes of 0.004–0.06 µg/mL [17], non-dermatophytes of 0.063–2.5 µg/mL [18], and yeast of 0.06–8 µg/mL [18].

All of them have proved to be effective in presenting transungual permeation. However, there is a need to improve the patient compliance with fewer repeat applications, meaning that topical nail formulations need to be improved. Nail lacquers, formulated with polymers that act as film former agents, could be an alternative to be used as new drug carriers [9].

In this study, the influence of polyurethanes (polymer) (PU) concentration on the properties of nail lacquer formulations were assessed and compared with a commercial formulation. The incorporation of two drugs (TH and CPX) validated the use of PU as versatile polymers in nail lacquer formulations. SEM, adhesion, drying time, viscosity and wettability measurements, antifungal activity, and nail studies, such as permeation and porosity, were performed to achieve these aims.

## 2. Material and Methods

### 2.1. Materials

Anhydrous ethanol, butyl acetate and ethyl acetate were acquired from Carlo Erba (Rueil-Malmaison, France). Terbinafine hydrochloride, manufactured by Uquifa México S.A., was kindly offered by Generis (Lisbon, Portugal). Tagat^®^ CH 60 (PEG-60 Hydrogenated Castor Oil) was purchased from Evonik (Essen, Germany). PU 19 was synthesized in laboratory of Superior Technical Institute, Universidade de Lisboa and the procedure was performed according a previous published work [19]. Ciclopirox was obtained from Fagron Iberica (Barcelona, Spain), Ony-Tec^®^ was from Laboratorio Medea, Reig Jofre (Barcelona, Spain). Methanol HPLC grade was purchased from Panreac (Castellar del Vallès, Spain). Triethanolamine was provided by Merck (Darmstadt, Germany).

### 2.2. Methods

#### 2.2.1. Preparation of Nail Lacquers

The PU 19, a polyurethane containing isophorone diisocyanate (IPDI) and polypropylene glycol (PPG) and d-isosorbide (6:1:5), was the polymer selected for preparing 5 different nail lacquer formulations containing 1% of drug (TH or CPX) [19]. Different polymer concentrations (10%, 15%, 20% and 25%) were employed. The formulations were prepared according to component described in Table 1. First, polyurethanes were fully solubilized in ethanol under stirring (200 rpm). Afterwards, the solvents and the drugs (TH or CPX) were added, separately, until complete dissolution. Placebos—formulations with no drug—were also prepared. A commercial formulation containing 8% (*w/v*) of CPX was also used as a control: Ony-Tec^®^ (composed by ethanol, water, ethyl acetate, hydroxipropyl chitosan and cetylstearyl alcohol).

#### 2.2.2. Preliminary *In Vitro* Release of Terbinafine from Nail Lacquers Containing Different Concentrations of PU

To evaluate the influence of different concentrations of polymer, an *in vitro* release study was performed using Franz diffusion cell apparatus through a hydrofilic membrane (Tuffryn^®^ Membrane, Pall corporation) (Portsmouth, UK), with a diffusion area of 1 cm^2^ for 6 h [20] and according a previous reported work [19]. The data obtained from *in vitro* release studies were fitted to different kinetic models:

(1) Higuchi model
$$F=\;K_{H}\;\times \;t^{1/2}$$
where, *K*_H_ is the Higuchi release constant.

(2) Korsmeyer-Peppas model
$$F=K_{\mathrm{KP}}\times t^{n}$$
where, *K*_KP_ is the release constant incorporating structural and geometric characteristics of the drug-dosage form and *n* is the diffusional exponent indicating the drug-release mechanism.

The determination of wettability by measurement of contact angle and the viscosity measurement were performed according a previous reported work [19].

#### 2.2.3. *In Vitro* Characterization Studies: SEM, Adhesion Tests and Determination of Antifungal Activity

The nail morphology and the adhesion determination were performed according to a previous reported work [19]. In addition, *Candida albicans* ATCC 10240 and *Aspergillus brasiliensis* ATCC 16404 were used for the determination of *in vitro* antifungal inhibitory activity of a terbinafine and cyclopirox nail lacquer formulations. The antifungal activity was also performed according to the research of Valdés et al. [19].

#### 2.2.4. Drying Time for Nail Lacquer Therapeutics

The nail lacquer must dry. A quantity of 0.25 ± 0.02 g of the nail lacquers were weighed and spread into a glass, to create a homogeneous film. The dryness time was measured with a chronometer. This methodology was adapted from ISO 2409:2013 [21].

#### 2.2.5. Final Formulations—*In Vitro* Drug Permeation and Porosity Nail Studies

##### Permeation Studies

The nail tips were obtained by cutting the free edge of the nail plate of a healthy volunteer (female 25 years old) after ethical approval and informed consent. The nail donation protocol was approved by the Ethic Committee of Galicia (2018/099). The samples had a minimum length of 5 mm. The nail tips were hydrated in 10 mL receptor solution (some in aqueous solution of 0.5% Tagat^®^ CH 60 and the other in phosphate buffer saline to which sodium azide) for 1 h. Ony-Tec^®^ was used as control. The nail tip samples were sandwiched between two cylindrical adapters made of polytetrafluoroethylene (PTFE) (Mecanizados del noroeste, Santiago de Compostela, Spain) with an o-shaped ring providing an effective diffusional area of 0.049 cm^2^. The set was placed between the donor and receptor chambers of vertical Franz-type diffusion cell (Vidrafoc, Barcelona, Spain) of the dorsal and ventral layers of the nails faced the donor and the receptor compartments, respectively. The donor chamber contained either 2 mL of one of the nail lacquers (Formulation A PU19-10% TH, formulation G PU19 10% CPX) and was covered with parafilm^®^. The receptor medium, 5.5 mL receptor volume, was aqueous solution of 0.5% Tagat^®^ CH 60 for Formulation A PU19-10% TH and phosphate buffer saline which sodium azide (30 mg/L) pH 7.4 [9] for Formulation G PU19-10% CPX and Ony-Tec^®^. The receptor compartments were at constant temperature by use of thermostatic (32 ± 0.5 °C) water. The sink conditions were assumed. Samples of 1000 µL were collected at predefined times (each day at the same time) and the same volume was replaced with fresh receptor solution maintained at the same temperature. The experiments were performed for 11 days and three replicates were made for each condition [9].

The cumulative amounts of drug diffused across the nail were normalized by the area and plotted versus time to estimate the pseudo-steady-state fluxes Jss by linear regression from the last linear portion of the profiles (between 3rd and 11th days) [9].

The amount of TH and CPX present in the nail plate at the end of penetration experiment was also determined by UV spectrophotometry (spectrophotometer diode array Hewlett Packard 8452A). The wavenumber for terbinafine was 283 nm [22] and for CPX 308 nm [9] after the drug extraction. The section of the nail exposed to the formulation was weighed and cut in small fragments that were transferred into a vial containing either 5 mL of receptor solution at 5% in methanol. Later, the prepared solutions were incubated and shaken for 4 days at room temperature to facilitate drug extraction. The CPX extraction method was adapted and validated from the literature [9,22] while the TH extraction method was validated for sensitivity, linearity and precision in the concentration range from 6.32 to 31.6 µg/mL (r^2^ = 0.9884, calibration curve standards were assayed together with every sample batch to account for inter day variability).

Standard and blank solutions were made up using phosphate buffer solution (PBS) buffer (pH 7.4) containing 30 mg/L of sodium azide and aqueous solution of 0.5% Tagat^®^ CH 60. The data were normalized to account for thickness of the nails.

##### Porosity Measurement

Untreated and treated nail samples were analyzed using a Micromeritics 9305 pore sizer (Norcross, GA, USA) fitted with a 3 mL powder penetrometer and working pressures in a 0.004–172.4 MPa range. Nail tips were used as described in Valdés et al. [19]. The nails were soaked in 10 mL of either water, solution ethanol 50% (*v/v*), Ony-Tec^®^ and Formulation A PU19-10% TH. Later, the samples were stored at room temperature for 24 h. After that time, the nails were removed from the solution, tested, dried, and the porosity determined. The amount of nail tip for test was approximately 0.6 g. The pore size data were used for modelling the porous structure of the samples as simulated porous networks using PoreXpert™ 1.3 software (Environmental and Fluid Modelling Group, University of Plymouth, UK) [23].

#### 2.2.6. Statistical Analysis

One-way analysis of variance when appropriate (ANOVA) and Tukey–Kramer post-hoc multiple comparison test was used to identify the significant differences between the groups and were performed using GraphPad PRISM^®^ 5 software. An α error of 5% was chosen to set the significance level unless stated otherwise.

## 3. Results and Discussion

### 3.1. Preliminary Tests for Select the Formulation with Better Condition to Release the Drugs

To confirm that the polymer concentration affects the drug release, an *in vitro* preliminary test was assessed just using the TH formulations. The results confirm that the highest the polymer concentration the less drug is released (Figure 1).

The Formulation A PU19-10% TH releases 67%, after 6 h, while the rest of formulation retains the terbinafine in the matrix for the same time of study. The Formulation D PU19-15% TH releases 16%, the Formulation E PU19-20% TH releases 10.6% and the Formulation F PU19-25% TH releases 6%. These results indicated that the increase of amount of polymer, decreases the release of terbinafine from the formulations.

Table 2 shows the fitting to the release profiles at the Kosmeyer and Peppas [24] and Higuchi kinetic [25]. Values of the *n* exponent of the formulation prepared with high polymeric concentrations (0.60–0.89) suggest that release is controlled by a mixture of relaxation of the polymeric film and the drug diffusion (case II transport). Nevertheless, the movement of the drug solute through the matrix system for PU19-10% is diffusion-controlled.

The results obtained demonstrated the nail lacquer formulation with lower amount of polymer release has an appropriate amount of drug for antifungal activity.

In the previous work [19] the relationship between the presence of water in the nail and the diffusion of drug was already mentioned and these results can be related to contact angle. When the contact angle is lower than 90°, this means that the nail lacquer will spread over a large area on the surface of the nail, releasing higher amounts of drug. In other words, the increase of hydrophilic properties of the formulation increases the drug’s release.

In the present research, different concentrations of polymer were used to prepare different formulations and its influence on contact angle were assessed. The results show that the increase in PU19 content results in higher contact angle values and hydrophobic films: for Formulation A PU19 10% a water contact angle of 45° ± 4° was obtained while for Formulation D PU19 15% the value was 56° ± 5° and for Formulation E PU19 20% and Formulation F PU19 25% the results were 66° ± 4° and 68° ± 11°, respectively (Figure 2). The contact angles results obtained for the 4 formulations are significantly different from the control (nail).

Table 3 shows the values of viscosity for the PU terbinafine nail lacquers at 32 °C to mimic nail lacquer application.

As expected, higher polymer concentration increases formulation viscosity.

Thus, to study the influence of drugs on the properties of nail lacquer formulations, CPX was incorporated in nail lacquer formulation containing 10% of PU19. All formulations containing TH were also studied.

### 3.2. SEM Analysis

The images obtained by SEM of nail lacquers show homogeneous films, which cover the surface of the nail as observed in Figure 3.

The inclusion of the drug (TH or CPX) had no influence on film thickness and on the microstructure of PU-based nail lacquers. However, Ony-Tec^®^ did not form a homogenous film in the surface of the nail. These measurements indicated that the application of the PU nail lacquers coated the roughness of the nail surface, making it smooth and uniform, while chitosan film from [9] was similar to the untreated nail with visible pores and cracks on the surface. 

### 3.3. Adhesion Test

The nail lacquer must adhere to nail. The duration of residence of the film on the nail plate is therefore critically important, because the film of nail lacquer acts as a drug depot, from which the drug can be continuously released and permeate into the nail. A film with a long residence time would need less-frequent lacquer application, which could in turn lead to increased patient compliance, improved treatment efficacy and reduced cost of treatment [26].

All the formulations presented adhesion to keratin of cow horn. The adhesion results were 1.8 ± 0.4, 1.8 ± 0.4, 1.6 ± 0.4, 1.8 ± 0.4, 0.2 ± 0.1 and 4.5 ± 0.6 for Formulation A PU19-10% TH, Formulation D PU19-15% TH, Formulation E PU19-20% TH, Formulation F PU19-25% TH, Formulation G PU19-10% CPX and Ony-Tec^®^, respectively.

The positive results of flaking symptoms were 57%, 86% and 75% for treatments with white tube, gray tube, and red tube, respectively.

The adhesion values in cow horn correspond to better results compared with the result of adhesion from nail lacquer formulated with methacrylates, where the cross-cut mean scores were between 4 and 5 [10].

By contrast, the nail lacquer that presents the higher flaking in the lattice pattern was formulation Ony-Tec^®^. The formulation Ony-Tec^®^ is a chitosan derivative’ hydrogel. Previously, a study by SEM showed a film with pores and cracks on the surface of the nail, and furthermore, hydroxypropyl chitosan is a water-soluble film former with affinity to keratin [27]. The results obtained allow the conclusion that the polyurethanes films present more affinity to keratin that Ony-Tec^®^.

The formulation with more adhesion to cow horn was the Formulation G PU 19-10% CPX, with a degree of flaking in the lattice pattern of 0.1, probably, because the formulation G PU19-10% CPX presented highest value ethyl acetate as solvent and permit more exposition of the hydrophilic group of polyurethane to keratin of cow horn [28].

### 3.4. Nail Lacquer’s Drying Time

For the compliance of the patient, the formulation must dry in the nail [28,29,30,31]. Table 4 shows the value of drying time of formulation.

The PU formulations present drying times from 9 to 16 min. The higher the concentration of the polymer, the longer the drying time of the nail lacquer. These results agree with the solvent contents because the higher the concentration of ethyl and butyl acetate the shorter the drying time. The Ony-Tec^®^ presents a drying time of 13 min, which is related to its composition. Ethanol is the main solvent of this formulation. According to these results, Formulation A PU19-10% TH, Formulation D PU19-15% TH and the Formulation G PU19-10% CPX should be selected.

### 3.5. Antifungal Activity

The antifungal activity of the drugs against *Candida albicans* ATCC 10240 and *Aspergillus brasiliensis* ATCC 16404 were performed. The results are shown in Table 5. The terbinafine presents antifungal activity against *C. albicans* [32] but the CPX is more effective against this yeast [18]. By contrast, the CPX presents lower activity against *A. brasiliensis* [18].

All the nail lacquer formulations present antifungal activity against the fungi of the study. The results obtained demonstrate the nail lacquer formulations with higher antifungal activity are the ones that have lower polymer concentration because they allow a higher release of the drug. All the control formulations presented an inhibition zone <6 mm.

The solution of TH formulation presented a higher inhibition zone against *C. albicans* probably because of the influence of NH of polyurethane in the pH of the medium that is the potential antifungal activity of the drug. The *in vitro* activity of terbinafine is pH-dependent and rises with increasing pH value [33]. The synergistic effect of the excipient and drug improves the antifungal activity of all nail lacquer formulations in contact with the fungus.

The Formulation G PU19-10% CPX presents one interesting value of antifungal activity in comparison with the CPX solution.

### 3.6. Nail Studies

#### Permeation Test

The amount of drug that permeates the nail in an animal model similar to that of man, for purposes of pre-clinical studies, is still under study and to date has not been established. The optional solution to be able to study the passage of the drug is the use of healthy human nails [4,34,35]. The results of the diffusion experiments carried out with CPX and TH into the nail are shown in Table 6. As was observed, in release experiments, the presence of polyurethane allows the diffusion of terbinafine trough the hydrophilic membrane and as expected the results of this study was similar.

The values of the flux and the drugs permeated thought the nail from Formulation A PU19-10% TH and from Formulation G PU19-10% CPX was higher than the observed in Ony-Tec^®^.

The comparison of the drug permeated at 72 and 264 h (3 and 11 days) shows that the higher diffusion occurs in the first 48 h after application, probably due to the presence of ethanol in the tested formulation that allows the rapid release of the drugs from the nail plate. The ethanol increased the drug solubility in the vehicle and permits high drug chemical activity values during the evaporation process promoting drug penetration. Once the ethanol is evaporated and the solid film is formed in the surface of the nail, the drug release slows down and so does the drug penetration. Furthermore, the small molecular size of both drugs allows the permeation through the nail (TH-291.4 g/mol and CPX-207.3 g/mol) [5].

Regarding the standard deviation in all the formulation testing, it is possible to conclude that the permeation process influences the thickness of nail if it is different from one individual to another and the composition of nail (percent of water, lipid, mineral) and drug-keratin interactions [36].

In other research where there was testing of the diffusion coefficient of amorolfine, terbinafine and caffeine, the standard deviation was significant. The author explains that the data were not corrected in terms of nail thickness; however, the number of replicated were *n* = 5–7 for caffeine, *n* = 11–13 for the amorolfine and *n* = 13–15 for terbinafine, which did not prevent the finding of equal and superior deviations from the determination [4].

The quantities of each drug were determined in the nail plate. Results are shown in Figure 4. The amount of CPX and TH determined in nail plate showed similar values; furthermore, the formulations-based polyurethane retain more drugs in the keratin structure than Ony-Tec^®^.

The amount of TH determined in the nail presented higher values (>0.004–1 µg/mL), and value of minimum inhibitory concentration of TH against *C. albicans* and *Aspergillus species* [18].

By contrast, the values of CPX determined in the nail are superior to the range 0.015–4 µg/mL corresponding to the minimum inhibitory concentration of the drug against *C. albicans* and *Aspergillus species*, respectively [18].

### 3.7. Porosity Study

The amount of CPX and TH diffused and retained in the nail plate showed similar values for PU19-based formulations. Thus, to evaluate the influence of the vehicle in nail plate hydration, a porosity study was performed.

The porosity of the nail is another parameter to take in count to explain changes in the diffusion of drugs due to modification on the hydration of nail plate. This hydration allows a drug flux from certain vehicles [23]. Indeed, an increase in nail plate hydration was found to increase the permeation of ketoconazole by three-fold and the diffusivity of water by more than 400-fold, when relative humidity increased from 15 to 100% [37]. One method useful for characterizing the internal microstructure of the nail is mercury intrusion porosimetry (MIP) in combination with PoreXpert™ software. The combination of this technique allows the obtaining of models useful for characterizing the permeability properties of the nails [23]. Figure 5 shows the values of porosity of the nail obtained by MIP before and after different treatment.

No significant changes on the microstructure of the nail were observed after the application of nail lacquers. The cumulative pore distribution curve of Ony-Tec^®^ seems to be relatively higher for lower pore diameter, but the model and the parameters characteristic of the model obtained by PoreXpert™ (Table 7) indicates similar porosity and connectivity regardless the formulation. The values of correlation of the models, near the unity, obtained in treated and untreated nail, indicates structures with high-level order caused by the well-delimited areas with high porosity at the top and bottom of the model and the low porosity structure in the middle of the model. Similar results were obtained by Nogueiras et al. [9] for healthy untreated nails using MIP plus PoreCore™ software.

These results indicated that PU19-10% placebo and Ony-Tec^®^ did not produce microstructural changes in the nail plaque, probably due to the presence of organic solvents and the absence (PU19-10%) or the low proportion of water (Ony-Tec^®^) used in the formulation of these nail lacquers. 

## 4. Conclusions

This study reports on the physic-chemical characterization of TH and CPX nail lacquer formulation-based polyurethane. The homogenous films obtained from the nail lacquer presented adhesion to nail. The viscosity, wettability and *in vitro* release studies reveal that the amount of PU19 influences the release profiles of the drug which is in accordance with the formulation’s antifungal activity.

All nail lacquer formulations showed antifungal activity against *C. albicans* and *A. brasiliensis*. The permeation profile of PU19-based formulations through the nail were higher than Ony-Tec^®^. Thus, it can be concluded that PU19-based nail lacquers are efficacious.

The use of polyurethane in nail formulation can be a promising strategy for the release lipophilic drugs.

## Figures and Tables

**Figure 1 pharmaceutics-10-00276-f001:**
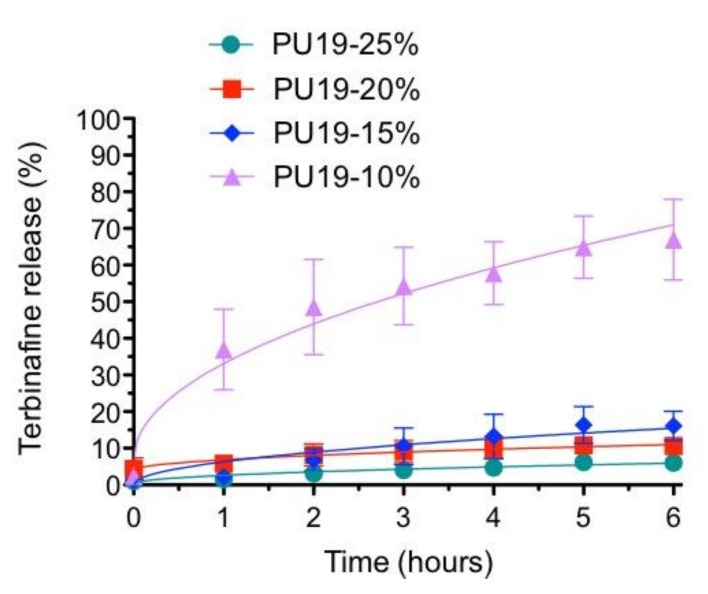
Release of TH from different formulations at 32 °C (mean ± SD, *n* = 6).

**Figure 2 pharmaceutics-10-00276-f002:**
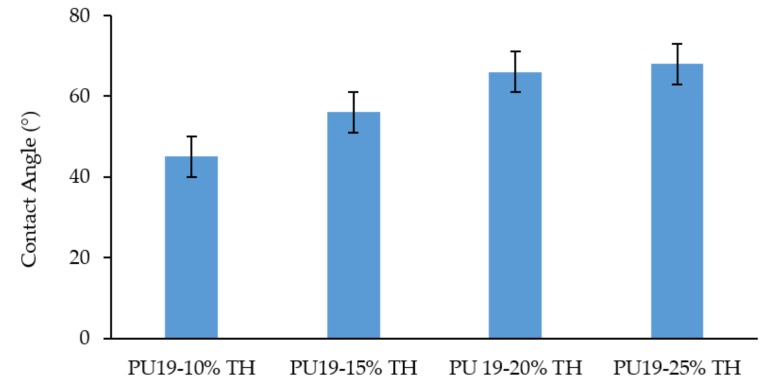
Water contact angle of PU terbinafine nail lacquers, nail as control, (mean ± SD, *n* = 6).

**Figure 3 pharmaceutics-10-00276-f003:**
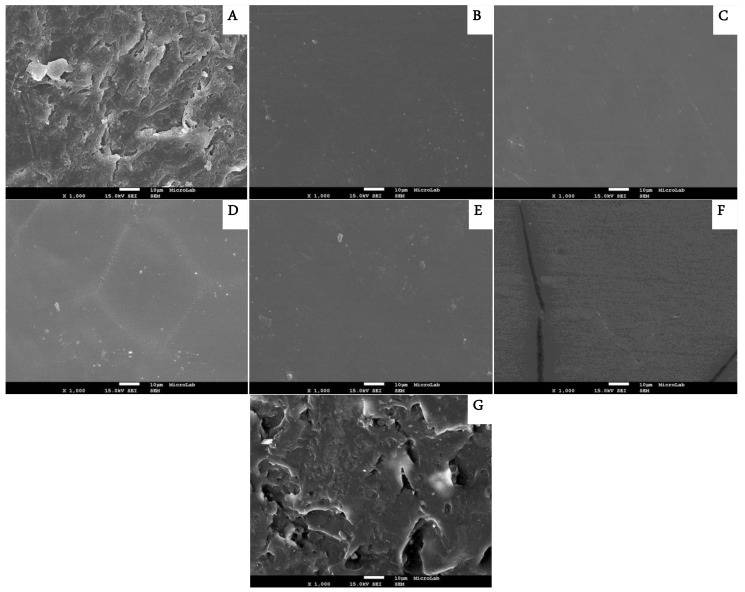
Scan Electron Micrograph with 1000× magnification. (**A**) dorsal surface of nail plate (**B**) film of Formulation A PU19-10% TH, (**C**) film of Formulation D PU19-15% TH, (**D**) film of Formulation E PU19-20% TH, (**E**) film of Formulation F PU19-25% TH, (**F**) film of Formulation G PU19-10% CPX and (**G**) Ony-Tec^®^. Scale bar (White line)—10 μm.

**Figure 4 pharmaceutics-10-00276-f004:**
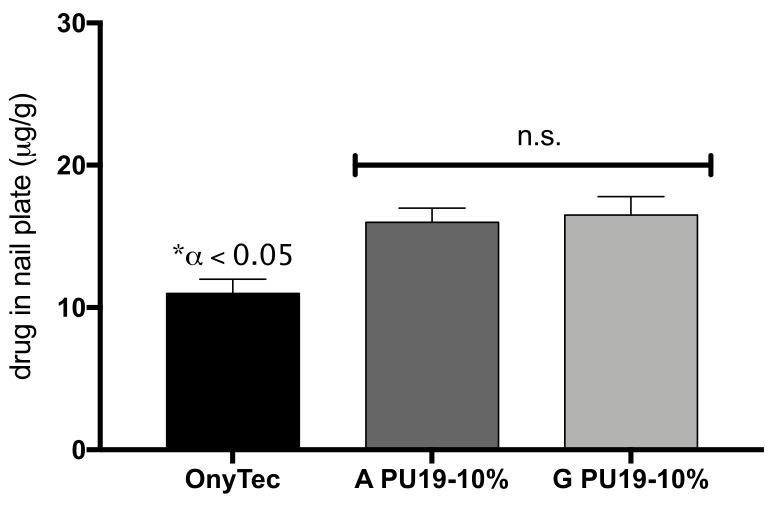
Amount of drug determined in nail plate after 11 days of experiments (mean ± SD).

**Figure 5 pharmaceutics-10-00276-f005:**
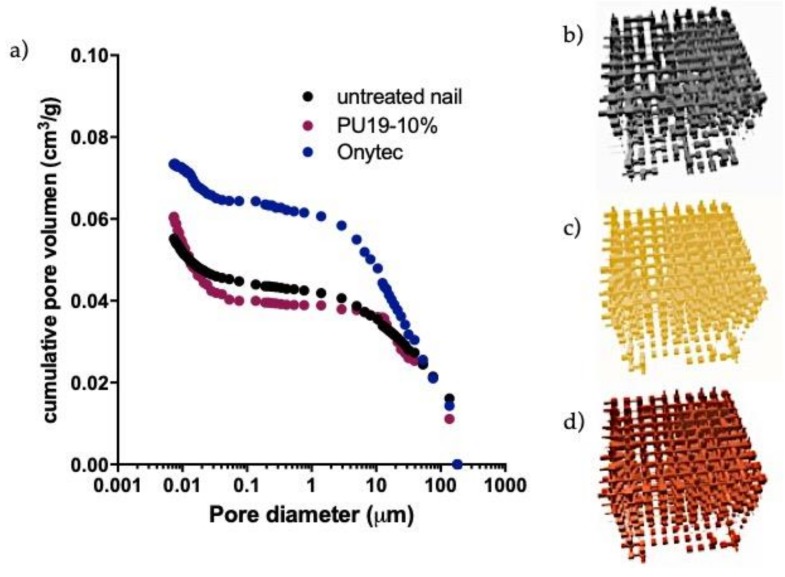
Cumulative curves of porosity obtained by MIP for treated and untreated for nail (**a**) and PoreXpert models of the microstructure of (**b**) untreated nail, (**c**) Ony-Tec^®^ treated nails and (**d**) PU19-10% treated nails. Cubes represent the pores and cylinders the connection between pores.

**Table 1 pharmaceutics-10-00276-t001:** Qualitative and quantitative composition of nail lacquers (%, *w/w*).

	Quantitative Composition (%, *w/w*)				
	Formulation A PU19-10% TH	Formulation D PU19-15% TH	Formulation E PU19-20% TH	Formulation F PU19-25% TH	Formulation G PU19-10% CPX
PU 19	10	15	20	25	10
Terbinafine HCl (TH)	1.0	1.0	1.0	1.0	-
Ciclopirox (CPX)	-	-	-	-	1.0
Ethyl acetate	7.8	7.2	6.8	6.3	17.8
Butyl acetate	10	9.6	9.0	8.5	-
Ethanol	71.2	67.2	63.2	59.2	71.2

**Table 2 pharmaceutics-10-00276-t002:** Fitting results of TH release profiles to Kosmeyer and Peppas [24] and Higuchi [25] kinetics.

	Formulation A PU19-10%	Formulation D PU19-15%	Formulation E PU19-20%	Formulation F PU19-25%
*Kosmeyer and Peppas*				
k	35.41	3.486	2.366	1.152
n	0.34	0.8924	0.6054	0.8864
r^2^	0.9979	0.9615	0.9588	0.9662
*Higuchi*				
k	26.12	6.32	2.872	2.304
r^2^	0.973	0.876	0.951	0.898

**Table 3 pharmaceutics-10-00276-t003:** Viscosity values of therapeutic nail lacquers (mean ± SD, *n* = 3).

Formulations	Viscosity (mPa.s)
Formulation A PU19-10% TH	2.62 ± 0.04
Formulation D PU19-15% TH	4.70 ± 0.06
Formulation E PU19-20% TH	8.80 ± 0.11
Formulation F PU19-25% TH	17.03 ± 0.07

**Table 4 pharmaceutics-10-00276-t004:** Nail lacquer’s drying time (*n* = 3).

Formulations	Time (min)
Formulation A PU19-10% TH	9
Formulation D PU19-15% TH	10
Formulation E PU19-20% TH	15
Formulation F PU19-25% TH	16
Formulation G PU19-10% CPX	7
Ony-Tec^®^	13

**Table 5 pharmaceutics-10-00276-t005:** Inhibition zone (mm) of all formulations in plate dish for *Candida albicans*, and *Aspergillus brasiliensis* (mean ± SD, *n* = 2).

Formulations	Inhibition Zone (mm)	
	*Candida albicans* ATCC 10240	*Aspergillus brasiliensis* ATCC 16404
Formulation A PU19 10% TH	38.4 ± 3.6	25.0 ± 0.4
Formulation D PU19 15% TH	29.7 ± 0.4	23.7 ± 0.6
Formulation E PU19 20% TH	29.0 ± 0.6	23.2 ± 0.1
Formulation F PU19 25% TH	26.2 ± 1.0	21.1 ± 1.7
Formulation G PU19 10% CPX	32.3 ± 1.3	26.7 ± 0.3
Solution TH (1%)	21.4 ± 1.8	32.0 ± 0.8
Solution CPX (1%)	29.0 ± 1.1	31.4 ± 0.4

**Table 6 pharmaceutics-10-00276-t006:** Delivery of CPX and TH across nail (mean ± SD, r^2^ = 0.85–0.95).

	**Co (mg/mL)**	**Acum_72h_ mg/cm^2^**	**Acum_264h_ µg/cm^2^**	**% Dose Delivered**	**J_app_ 48–264 h**
Ony-Tec^®^	80	0.19 ± 0.17	0.29 ± 0.16	0.01 ± 0.005	14.1 ± 2.68
Formulation G PU19-10% CPX	10	0.51 ± 0.35	0.87 ± 0.50	0.21 ± 0.12	50.5 ± 4.12
Formulation A PU19-10% TH	10	0.48 ± 0.03	0.80 ± 0.11	0.20 ± 0.03	36.5 ± 3.69

**Table 7 pharmaceutics-10-00276-t007:** Main parameters of the models obtained from porosity data using PoreXpert™.

	Porosity (%)	Correlation	Connectivity	Water Permeability (mD)
Untreated nails	6.83	0.795	5.02	1.5775 × 10^−6^
Ony-Tec^®^	6.75	0.796	5.02	1.5417 × 10^−6^
Formulation A PU19 10% TH	6.75	0.795	5.02	1.5775 × 10^−6^

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
