# Peer review of "Polyurethanes as New Excipients in Nail Therapeutics"

_pharmaceutics, 2018, doi:10.3390/pharmaceutics10040276_

Round 1
Reviewer 1 Report
The manuscript by Gregorà Valdes et al. described a research in using polyurethane coatings for onychomycosis. The manuscript is well written and clearly demonstrated the findings to the readers. The research manuscript fits in the Journal’s scope and contributes to the knowledge of current field.
My suggestion to the research work is to provide additional data in water contact angle of the blank and drug-loaded polyurethane coatings. This will help to support the drug release data.
Author Response
1. My suggestion to the research work is to provide additional data in water contact angle of the blank and drug-loaded polyurethane coatings. This will help to support the drug release data.
Answer: The comment of the reviewer is pertinent and the additional data requested has been made in the revised version of the manuscript.
Reviewer 2 Report
The manuscript by Valdes et al describes the assessment of using polyurethane as the basis of nail lacquer and its comparison with the commercial formulation.
(1) What are the justification of selecting polyurethane? The commercial formulation, Ony-Tec, uses chitosan which can be cheaply produced naturally in a large scale.
(2) In Table 1, the total percentage of formulation A is not 100%. Please revise. Also, why the formulation G does not contain butyl acetate as in all of the other formulations?
(3) The authors do mention the role of hydrophilic polymer and increased viscosity influence the matrix dissolution and erosion (lines 182-186). But it is not clear in the text whether polyurethane is hydrophilic and its increased concentration causes increased viscosity as well. The authors only used hydroxyethyl cellulose as an example to explain polyurethane. The effect of polyurethane concentration on viscosity should also be demonstrated and explained before the release-kinetic study.
(4) The equations Kosmeyer and Peppas kinetics and Higuchi kinetics were used to explain the release profiles of the drugs, but there are no equation presented on the manuscript.
(5) The scale bar in Figure 2 is not clear. Please revise. Also, what are the advantages and disadvantages of having homogeneous film covered the surfaces?
(6) In Table 3, Formulations A and G contain the same amount of polyurethane i.e. 10% (w/w), but different drugs. Please explain why the drying time for formulation G is two minute faster than formulation A.
(7) As aforementioned, the viscosity values in Section 3.5 should be mentioned earlier before Section 3.1. Moreover, the explanation in Section 3.1 should also connect with the work in Section 3.5 to make it cohesive.
(8) Are there justifications as to why antifungal-activity studies were performed instead of antibacterial-activity studies? Why were C. albicans and A. brasiliensis selected to perform such studies instead of other organisms? What properties in those two organisms that cause different inhibition zones after the treatment using the same formulation?
(9) Was the drug content in each formulation in Table 5 the same? The control formulation, Ony-Tec, showed larger inhibition zone than that of the formulations developed in this manuscript. Does that mean that the new formulation has worse performance than the control?
Author Response
1. What are the justification of selecting polyurethane? The commercial formulation, Ony-Tec®, uses chitosan which can be cheaply produced naturally in a large scale.
Answer: The polyurethanes are copolymers which have been used for biomedical applications for over three decades, however they have never been used in therapeutic nail lacquers. The polyurethane system is being used for sustained and controlled delivery of various dosage forms. These systems are based on a physical combination of the drug with polymers and kinetics of drug release is generally controlled by the diffusion phenomena through the polymer. The fact of being a synthetic polymer offers advantages to the formulation made, since with a small change in the composition of the monomers that take part of its structure, it can become a new polyurethane with different properties.
This meant that a research can develop different polyurethanes according to the decided use, such as: increase the release of different drugs, terbinafine hydrochloride and cyclopirox olamine or improve adhesion or become a polyurethane compatible with a tissue cell: in this case, with HaCaT cell.
On the other hand, chitosan is a natural polycationic linear polysaccharide derived from chitin, produced cheaply on a large scale in a natural way. The low solubility of chitosan in neutral and alkaline solution limits its application. The lower adherence to some tissues, such as nails and a lower solubility in certain volatile polar solvents, affects the use in therapeutic nail lacquer. Nevertheless, chemical modification into composites or hydrogels brings to it new functional properties for different applications. Chitosans are recognized as versatile biomaterials because of their non-toxicity, low allergenicity, biocompatibility and biodegradability. Taking into account that the formulation that exists in the market (OnyTec®, has limitations in its effectiveness and one of the componet is hydroxipropyl chitosan (semi-synthetic polymer). Polyurethane could be studied as a new excipient in the treatment of nails.
2. In Table 1, the total percentage of formulation A is not 100%. Please revise. Also, why the formulation G does not contain butyl acetate as in all of the other formulations?
Answer: Actually, there was a mistake: Instead of "17.8", it should be "7.8". Error corrected in the revised manuscript. The formulation G does not contain butyl acetate as in all of the other formulation because contains Ciclopirox olamine that is not soluble in butyl acetate.
3. The authors do mention the role of hydrophilic polymer and increased viscosity influence the matrix dissolution and erosion (lines 182-186). But it is not clear in the text whether polyurethane is hydrophilic and its increased concentration causes increased viscosity as well. The authors only used hydroxyethyl cellulose as an example to explain polyurethane. The effect of polyurethane concentration on viscosity should also be demonstrated and explained before the release-kinetic study.
Answer: The comment of the reviewer is pertinent and it was corrected in the revised manuscript.
4. The equations Kosmeyer and Peppas kinetics and Higuchi kinetics were used to explain the release profiles of the drugs, but there are no equations presented on the manuscript.
Answer: The clarification requested has been made in the revised version of the manuscript and the equations were now included.
5. The scale bar in Figure 2 is not clear. Please revise. Also, what are the advantages and disadvantages of having homogeneous film covered the surfaces?
Answer: The quality of scale bar was improved. The homogeneous film covered the surfaces, in this case the surface of the nail is one advantage in way to control the release of the drug in all the surface of nail. The homogeneous films allow controlled the diffusion of the drug from the polymeric matrix and improve the bioavailability of the delivery systems. If not exist one homogeneous film covering the surfaces, in this case of the nail, the drug not be release with control in all the surface affected by the onychomycosis and this is a disadvantages for the treatment of the disease.
6. In Table 3, Formulations A and G contain the same amount of polyurethane i.e. 10% (w/w), but different drugs. Please explain why the drying time for formulation G is two minute faster than formulation A.
Answer: The drying time for formulation G is two minute faster than formulation A, due to the presence of butyl acetate.
7. As aforementioned, the viscosity values in Section 3.5 should be mentioned earlier before Section 3.1. Moreover, the explanation in Section 3.1 should also connect with the work in Section 3.5 to make it cohesive.
Answer: The comment of the reviewer is pertinent, and it was corrected in the revised manuscript.
8. Are there justifications as to why antifungal-activity studies were performed instead of antibacterial-activity studies? Why were C. albicans and A. brasiliensis selected to perform such studies instead of other organisms? What properties in those two organisms that cause different inhibition zones after the treatment using the same formulation?
Answer: The antifungal-activity studies were performed instead of antibacterial-activity studies because the selected model drugs (terbinafine hydrochloride and ciclopirox olamine) are antifungal drugs. Onychomycosis is a nail fungal infection caused by dermatophytes, non-dermatophytes and yeast species [Ref:1]. Candida species have a high incidence in fingernail infection, present in as many incidences of as 75% of cases, and are more prevalent than dermatophytes [Ref:2]. For all the mention previously in the text was selected Candida albicans and Aspergillus brasiliensis to perform antifungal-activity studies.
The ciclopirox olalamine is used in nail topical formulations and it have been marketed worldwide for about 25 years and they are available in different pharmaceutical preparations. The present a minimum inhibitory concentration value for C. albicans and Aspergillus species between 0.13-4 µg/mL [Ref:14,15].
On the other hand, Terbinafine hydrochloride is an allylamine derivative, represents the most effective antimycotic drug, presenting a minimum inhibitory concentration against dermatophytes of 0.004-0.06 µg/mL [17], non-dermatophytes (Aspergillus spp) of 0.063-2.5 µg/mL [Ref:18] and yeast (Candida spp) of 0.06-8 µg/mL [Ref:18].
The enzymatic systems of each microorganism defined different properties in these two different microorganisms (C. albicans (yeast) and A. brasiliensis (non-dermatophytes)), which cause different zones of inhibition after treatment with the same formulation.
9. Was the drug content in each formulation in Table 5 the same? The control formulation, Ony-Tec, showed larger inhibition zone than that of the formulations developed in this manuscript. Does that mean that the new formulation has worse performance than the control?
Answer: The drug content in each formulation in Table 5 is not the same. The formulations prepared with polyurethane have only 1% of model drugs (terbinafine hydroclhoride and ciclopirox olamine), on the other hand, Ony-Tec® has 8% of ciclopirox olamine. The main objective of this study was to test the antifungal activities of the formulations prepared with polyurethane, despite being very different in concentration compared to the commercial product (Ony-Tec®). The differences between the concentration of Ony-Tec® and the formulations developed can create a misinterpretation and imply that they are not effective. Thus, the results of Ony-Tec® were delete. However, the authors used as positive controls the solutions of model drugs.